# Present and Future Losses of Storage in Large Reservoirs Due to Sedimentation: A Country-Wise Global Assessment

**Duminda Perera** [1,2,3,*] , **Spencer Williams** [4] **and Vladimir Smakhtin** [1]

[1] United Nations University Institute for Water, Environment and Health, Hamilton, ON L8P 0A1, Canada
[2] Department of Civil Engineering, Faculty of Engineering, University of Ottawa, Ottawa, ON K1N 6N5, Canada
[3] School of Earth, Environment and Society, McMaster University, Hamilton, ON L8S 4L8, Canada
[4] Faculty of Law, McGill University, Montreal, QC H3A 0G4, Canada
[*] Correspondence: duminda.perera@unu.edu; Tel.: +1-905-667-5483

**Abstract:** Reservoir sedimentation is often seen as a site-specific process and is usually assessed at an individual reservoir level. At the same time, it takes place everywhere in the world. However, estimates of storage losses globally are largely lacking. In this study, earlier proposed estimates of sedimentation rates are applied, for the first time, to 47,403 large dams in 150 countries to estimate cumulative reservoir storage losses at country, regional, and global scales. These losses are estimated for the time horizons of 2022, 2030, and 2050. It is shown that 6316 billion $m^3$ of initial global storage in these dams will decline to 4665 billion $m^3$ causing a 26% storage loss by 2050. By now, major regions of the world have already lost 13–19% of their initially available water storage. Asia-Pacific and African regions will likely experience relatively smaller storage losses in the next 25+ years compared to the Americas or Europe. On a country level, Seychelles, Japan, Ireland, Panama, and the United Kingdom will experience the highest water storage losses by 2050, ranging between 35% and 50%. In contrast, Bhutan, Cambodia, Ethiopia, Guinea, and Niger will be the five least affected countries losing less than 15% of storage by 2050. The decrease in the available storage by 2050 in all countries and regions will challenge many aspects of national economies, including irrigation, power generation, and water supply. The newly built dams will not be able to offset storage losses to sedimentation. The paper is an alert to this creeping global water challenge with potentially significant development implications.

**Keywords:** sedimentation; large dams; dam storage; dam reservoirs; storage losses

## 1. Introduction

Water storage infrastructure is critical for development. Large dams and reservoirs provide hydroelectricity, flood control, irrigation, and drinking water and often perform multiple functions simultaneously. A dam is considered "large" if it is higher than 15 m or between 5 and 15 m high but impounds over 3 million $m^3$ [1]. Construction of large dams peaked in the 1960 and 1970s, and today there are nearly 60,000 such dams worldwide (ICOLD, 2020) [1]. Thousands of them have aged considerably since then, now facing a higher risk of failure or becoming less effective [2].

One significant contributor to this decline in performance is reservoir sedimentation, resulting from dams impeding rivers' natural sediment transport process [3]. Reservoir sedimentation reduces functional storage capacity due to sediment accumulation and deposition [4]. The awareness of reservoir sedimentation predates the explosive growth in dam building in the 20th century. However, it has often been ignored—sometimes until a reservoir fills with sediment and becomes a liability to owners or downstream residents [5]. By now, reservoir sedimentation has become a significant challenge to global water storage infrastructure that must be addressed with similarly global and coherent action. It was foreseen by some over two decades ago that, in the 21st century, it will be necessary to

focus on combating sedimentation to extend the life of existing infrastructure [6]. Today, "reservoir design and management without a long-term sediment management strategy is not a sustainable approach, and no longer represents an engineering best practice" [7].

Sedimentation endangers the sustainability of the future water supply as it affects a reservoir upstream, a river downstream, and a dam infrastructure [8]. Sediment accumulation upstream stimulates flooding, while the reduction in sedimentation downstream causes erosion that impacts wildlife habitats and coastal populations. Abrasive sediments can damage turbines and other dam components and mechanisms, decreasing their efficiency and increasing maintenance costs [9].

Sedimentation is a common phenomenon that occurs, to varying degrees, in all reservoirs—small or large. Accumulation of sediments obviously decreases a reservoir's capacity over the years and determines a reservoir's life expectancy. Net global reservoir capacity is believed to have peaked in 2006 [10]. Since then, storage losses globally appear to have exceeded growth in storage capacity associated with new large dams' construction (which, in turn, has dropped significantly, e.g., Perera et al. [2]). Compared to the peak global per capita reservoir storage in the latter half of the 20th century, current storage capacity has decreased and become equivalent to pre-peak levels of the early 1960s, mainly due to sedimentation [11].

A large and growing body of literature examines the sedimentation of individual reservoirs or sets of reservoirs in a particular administrative or geographic area and provides numerical estimates of storage losses. Soler-López and Licha-Soler [12] conducted a bathymetric survey of Lago Loíza reservoir formed by the Carraizo Dam in Puerto Rico and estimated that annual storage loss from 2004 to 2009 was around 1.2% of reservoir capacity. Garg et al. [13] assessed storage losses in the Tehri Reservoir in India since 2005 using multi-temporal satellite images, and Mupfiga et al. [14] applied a similar approach to the Tuli-Makwe Dam in Zimbabwe. While in the former case, storage loss was relatively small (just over 1% of the initial capacity over the period), in the latter, over 40% of the initial capacity was lost after 50 years of sedimentation. Rahmani et al. [15] used a combination of acoustic bathymetric surveys' data and empirical modeling to illustrate that annual storage capacity loss for 24 federally operated large dams in Kansas, USA, over the period 1998–2014 ranged from 1.2% to 45.4% of their initial capacity, with an average loss of around 18%. Minear and Kondolf [16] developed a spreadsheet-based model that iteratively calculates sediment yield, accounting for trapping by upstream reservoirs and changing trap efficiency with time-to evaluate storage losses in California and concluded that over 190 reservoirs in the state had their capacity reduced below 50% of the original capacities, and 120 reservoirs were below 25%. Bilal et al. [17] used a numerical model to calculate the storage capacity loss of the Sakuma reservoir in Japan between 1956 and 2004 and to project the losses in the future. They estimated that the reservoir would lose around 44% of its initial capacity by 2040. Jabbar and Yadav [18] developed a reservoir capacity loss model based on sediment rating curves and applied it to the Shetrunji reservoir in India, illustrating that around 1% of gross capacity was lost annually during 1996–2000; the results were in good agreement with the data from hydrographic surveys.

Attempts to estimate some "global" annual rate of storage losses, while somewhat uncertain, seem to agree that it ranges between 0.5% and 1% of initial reservoir capacity [19,20]. However, the examples above suggest that reservoir sedimentation rates and associated storage losses are site-specific and vary significantly between regions. Perhaps the only comprehensive attempt to date to estimate reservoir sedimentation rates/storage losses at the global scale is that of Wisser et al. [10], who suggested several different reservoir storage loss rates and applied them to the selected 6399 reservoirs from the Global Reservoir and Dam (GRanD) database. GRanD includes 6862 georeferenced dams with a total capacity of 6197 km$^3$ and their associated reservoirs' attributes [21].

The current study attempts to estimate a global variation of storage losses of large dams due to sedimentation in much more detail. While it utilizes the storage loss rates suggested by Wisser et al. [10], they are applied in this study to the much larger and most

comprehensive global dataset of large dams available at present—that of the International Commission on Large Dams (ICOLD). Additionally, the study makes estimates of storage losses by country and ranks countries by the magnitude of their storage losses to sedimentation. A country-wise approach is essential, as the problems of reservoir sedimentation and overall future water storage development are dealt with by each nation individually. The study also assesses storage losses for several time horizons—2022, 2030, and 2050. This examines the relative urgency of addressing the issue of storage losses in different countries and regions of the world.

## 2. Methodology

Storage loss due to sedimentation with time is determined by a certain loss rate and initial reservoir storage capacity [10]:

$$C_t = \max(0,\ C_0 - \frac{LR}{100}C_0 t) \tag{1}$$

where $C_t$ is the storage at time $t$ (year); $C_0$ is the initial reservoir capacity at the time of construction (m$^3$); and LR is an annual loss rate (% of reservoir capacity).

Wisser et al. [10] used successive bathymetry survey data from 1024 dams from the USA's Reservoir Sedimentation Database (RESSED) and data from other 191 reservoirs from various other countries to estimate several annual LRs: minimum, average, median, capacity-weighted, and maximum. These five global LRs rates also differed between the initial capacity of the reservoirs. They were estimated for reservoirs with a capacity larger than 1 million m$^3$ and reservoirs under 1 million m$^3$ (Table 1).

**Table 1.** Statistics of observed annual reservoir loss rates (% of capacity) as summarized by Wisser et al. [10].

| Capacity (C) | Minimum LR. | Average LR. | Median LR. | * Capacity-Weighted Mean LR. | Maximum LR |
|---|---|---|---|---|---|
| C < 1 million m$^3$ | 0.0017 | 1.73 | 0.70 | 0.89 | 36 |
| C > 1 million m$^3$ | 0.0041 | 0.76 | 0.35 | 0.66 | 15 |

* capacity-weighted mean loss rate–Loss rate weighted by the initial capacity of the reservoirs.

In the current study, the reservoir storage losses due to sedimentation were estimated by applying some of the LRs listed in Table 1 to the large dams from the ICOLD database that contains data on nearly 59,000 dams. Wang et al. [22] attempted to improve and refine the ICOLD database for USA and Canada by adding georeferenced attributes and cleaning the database for repetitions. These enhanced data were used in the current study for the USA and Canada, and the rest of the data were directly extracted from the ICOLD database.

Besides LRs, the required parameters for this study are initial storage capacity ($C_0$) and construction completion year (i.e., the first year of the exploitation of the dam). A review of the ICOLD database reveals that these parameters are available for a subset of 47,403 dams out of nearly 59,000 dams on the ICOLD database. This is over 85% of all large dams included in the database. At the regional level, Asia-Pacific—28,045 (80% of total registered), Africa—2349 (91% of total registered), Europe—6651 (95% of total registered), and the Americas (North, Central, and South)—10,358 dams (93% of total registered) were included in the storage loss calculation. Overall, 150 countries were included in this study, 45 being from the Asia-Pacific region, 44 from Africa, 42 from Europe, and 19 from the Americas.

At a country or regional level, initial total storage is the sum of all individual initial (i.e., at the first year of the dam exploitation) capacities of large dams in that country/region, irrespective of the year of construction. Storage losses due to sedimentation were estimated for the present (2022) and future time horizons: years 2030 and 2050.

The maximum and minimum LRs were not considered as these rates are extreme cases unlikely to prevail annually. An attempt was further made to establish which of the three remaining LRs is the most likely by comparing the storage loss estimates of Wisser et al. [10] with documented ones found in various literature sources from around the globe. Altogether, 247 such recorded storage loss data were identified (i.e., India (211) [23], USA (24) [24], Japan (8) [17,25], Iran (1) [26], Iraq (1) [27], Lesotho (1) [28], and Puerto Rico (1) [12]). Table 2 describes the accuracy assessment for the Wisser et al. [10] proposed LRs-based storages and published storages using Root Mean Square Error (RMSE) and Nash–Sutcliffe coefficient of efficiency (NSE).

**Table 2.** Error Estimation Statistics.

| LRs | RMSE | NSE |
|---|---|---|
| Average | 346.4 | 0.94 |
| Median | 103.2 | 0.98 |
| C-weighted Mean | 277.2 | 0.96 |

The median LR was found to correlate best with documented estimates in terms of RMS and NSE. Accordingly, while estimates of storage losses have been completed using all three LRs (median, average, and capacity-weighted) as possible sedimentation scenarios, the results in the next section are presented for what is perceived as the most likely scenario—the median LR.

## 3. Results and Discussion

Tables 3 and 4 summarize the results for large reservoirs' storage losses and storage that will still be available due to these losses for various time horizons and by the major geographical regions. For illustrative purposes, all North, Central, and South American countries were combined in one region—the Americas. All estimates presented are completed with the "median LR" only. It is obvious that "available storage" is an inverse of "storage loss", and hence in most subsequent illustrations, only the "available loss" variable was used.

**Table 3.** Estimates of available storage in 2022, 2030, and 2050.

| Region | Number of Countries and (Dams) Considered | Initial Total Storage Billion, m$^3$ | Available Storage-2022 | | Available Storage-2030 | | Available Storage-2050 | |
|---|---|---|---|---|---|---|---|---|
| | | | Billion, m$^3$ | (%) | Billion, m$^3$ | (%) | Billion, m$^3$ | (%) |
| Americas | 19 (10,358) | 2810 | 2289 | 81% | 2210 | 79% | 2014 | 72% |
| Europe | 42 (6651) | 895 | 730 | 82% | 704 | 46% | 642 | 72% |
| Africa | 44 (2349) | 702 | 599 | 85% | 579 | 83% | 530 | 79% |
| Asia-Pacific | 45 (28,045) | 1909 | 1664 | 88% | 1611 | 85% | 1479 | 77% |
| Total | 150 (47,403) | 6316 | 5282 | 84% | 5104 | 81% | 4665 | 74% |

**Table 4.** Estimates of storage loss in 2022, 2030, and 2050.

| Region | Number of Countries and (Dams) Considered | Initial Total Storage Billion, m$^3$ | Storage Loss-2022 | | Storage Loss-2030 | | Storage Loss-2050 | |
|---|---|---|---|---|---|---|---|---|
| | | | Billion, m$^3$ | (%) | Billion, m$^3$ | (%) | Billion, m$^3$ | (%) |
| Americas | 19 (10,358) | 2810 | 521 | 19% | 600 | 21% | 797 | 28% |
| Europe | 42 (6651) | 895 | 166 | 19% | 191 | 21% | 254 | 28% |
| Africa | 44 (2349) | 702 | 103 | 15% | 123 | 17% | 172 | 24% |
| Asia-Pacific | 45 (28,045) | 1909 | 245 | 13% | 299 | 16% | 432 | 23% |
| Total | 150 (47,403) | 6316 | 1035 | 16% | 1212 | 19% | 1655 | 26% |

The regionwide storage loss percentages for 2022 range between 13% and 19%; for 2030, between 16% and 21%; and for 2050, between 23% and 28%. Globally, storage losses

by now have reached 16% of the initial storage capacity. By 2050, storage losses in Asia-Pacific are estimated to be the lowest (23%), while in the Americas and Europe they are estimated to be the highest (28%). Globally, an additional 10% of storage loss will likely occur from 2022 (16%) to 2050 (26%); in other regions, 9%. Annually, the globally averaged losses amount to approximately 0.36% of initial global storage capacity.

Figure 1 illustrates that Asia (except for Japan), Africa, and South America show a below 10% (relatively low) level of storage loss at present (2022), and it will increase consistently to between 10% and 25% by 2030. When 2050 is reached, except for a few countries in Asia, Africa, and South America, all others approach the 25% storage loss level. The primary reason for this could be the younger dams in these regions due to continuous dam building in the recent past (Perera et al., 2021) [2]. Europe, North America, Australia, and Japan show an extensive loss of storage capacities (Figure 1). Having large and aged dams expand the sediment accumulation, resulting in a significant loss of storage.

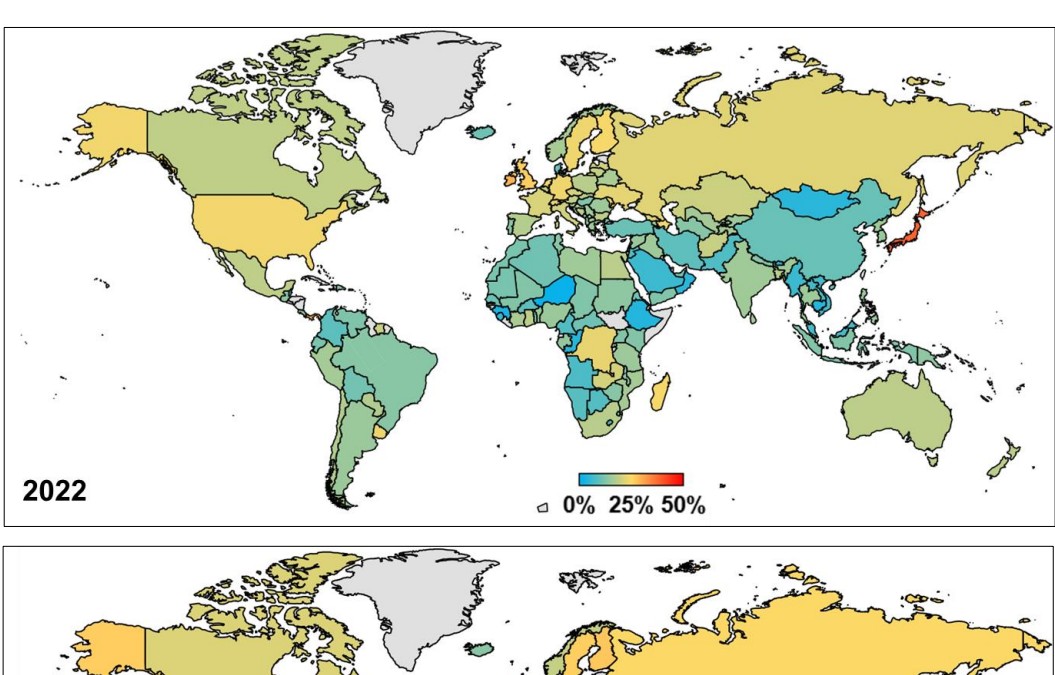

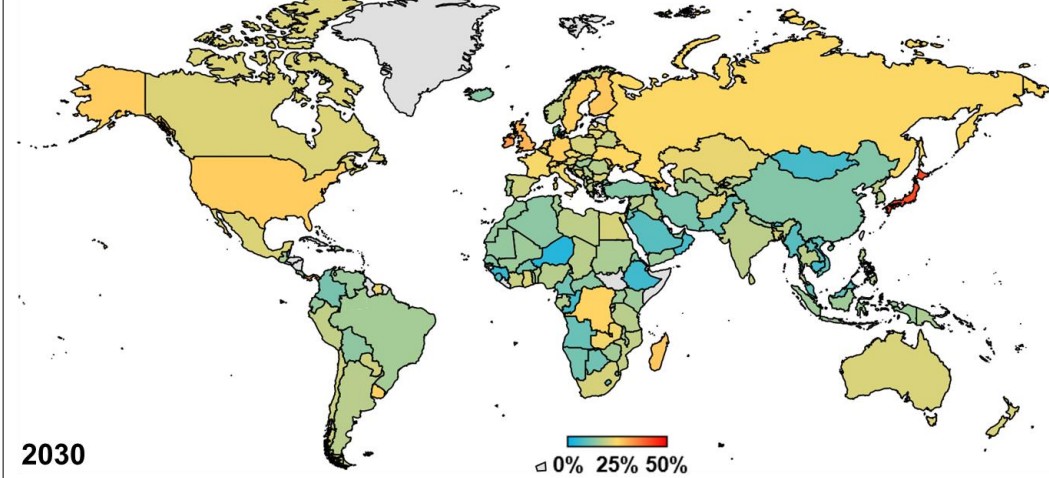

**Figure 1.** *Cont.*

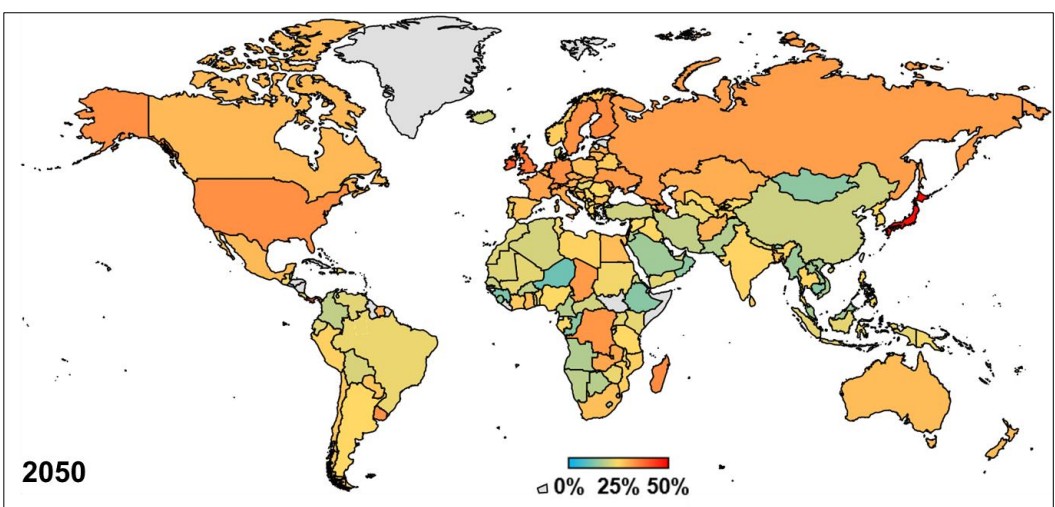

**Figure 1.** Estimated storage loss (%) by 2022, 2030, and 2050.

### 3.1. Americas

In this assessment, the Americas include 19 countries and 10,358 large dams, with a cumulative initial storage capacity of 2810 billion m$^3$. This is estimated to decline to 2014 billion m$^3$ by 2050 (a decrease of 28%). Although the USA, Mexico, Brazil, and Canada are the top four nations with large dams in the region [2], Panama appears to be the country with the highest storage loss (38%; Figure 2). A total of 21 dams in Panama were included in the assessment, with their initial storage of 9.5 billion m$^3$. The projected reduction is to 5.9 billion m$^3$ by 2050.

The USA is in second place (Figure 2), having a 34% loss of storage by 2050 in its 7469 large dams considered in the study. The 10% storage loss in 28 years from 2022 will make the storage available in 2050 equal to 580 billion m$^3$. Tullos et al. [29] stated that the USA's absolute storage capacity was reduced between 10% and 35% due to sedimentation.

Uruguay, Suriname, and Canada are estimated to lose 33%, 30%, and 29% of their initial storage capacity, respectively, by 2050. At the "tail end" of storage losses in the Americas are Bolivia (21%), Colombia (19%), and Belize (16%). Currently (2022), only Panama in the Americas has lost over 25% of initial storage. By 2030, USA and Uruguay will reach the same level of losses, and by 2050, 10 out of 19 countries in the Americas will lose over 25% of their initial storage.

Brazil, the second after the USA in the Americas in terms of large dam numbers, will lose 23% of its initial storage of 600 billion m$^3$ by 2050. Between 2022 and 2050, the country's storage will decline by 59 billion m$^3$, i.e., approximately 0.35% of initial storage annually. De Araujo et al. [30] observed that Brazil's storage capacity decreased by 0.2% annually due to sedimentation.

Based on the 57 large Peruvian dams included in this study, it is estimated that, by 2050, the country will likely lose 26% of its initial storage due to sedimentation. Cabrera and Gambini [31] examined the Poechos reservoir—the largest in Peru—which was completed in 1977. The reservoir volume was already less than half of its initial capacity at the time of their study, and they estimated that by 2035 the reservoir could be filled with sediments. If the median LR is used for the Poechos reservoir, the estimate results in only 20% storage loss by 2035. This example suggests that a median LR used in the current study (at least in some cases) is relatively modest and that actual losses due to sedimentation in certain regions may be much more significant.

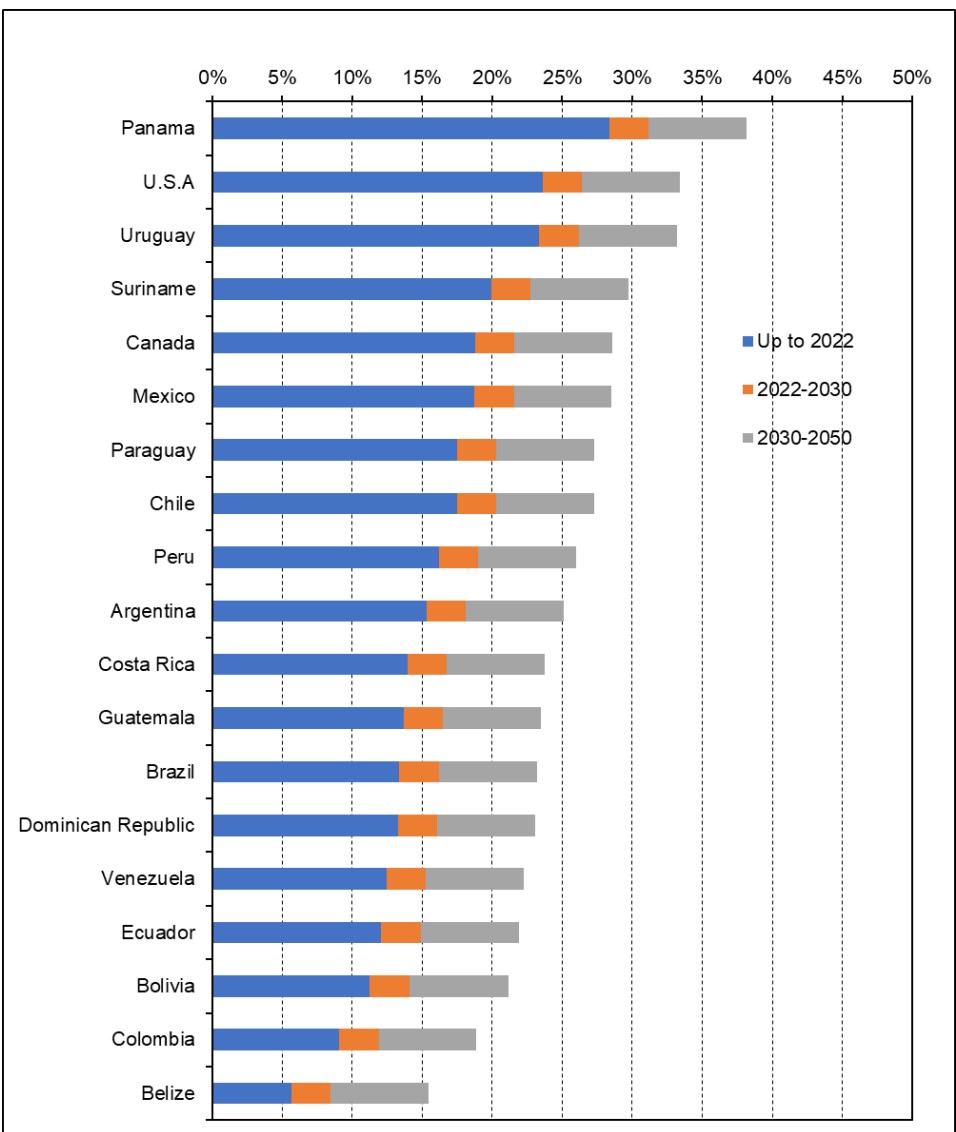

**Figure 2.** Estimated reservoir storage loss (%) by country in the Americas.

### 3.2. Europe

For Europe, 6651 large dams from 42 countries were considered in the study. The total initial storage of those dams was 895 billion m$^3$; however, Europe already lost 19% of it by 2022 and will lose up to 21% by 2030 and 28% by 2050. Europe "shares" the highest storage loss percentage by 2050 with the Americas. Among the 42 countries, 33 (~78%) will likely lose over 25% of initial storage by 2050 (Figure 3)—doubling its 2022 number and a significant number of such countries compared to other regions. A likely primary reason for this is that most dams are aged and have accumulated already (and will accumulate) more sediments. Ireland has the highest level of storage loss at all time horizons (e.g., 39% by 2050), followed by the UK, Finland, the Netherlands, and Moldova. Denmark has the lowest storage loss among the 42 countries. It would be 20% of its initial storage by 2050. Turkey, Iceland, Hungary, and Cyprus appear to be the other few least-impacted countries in Europe.

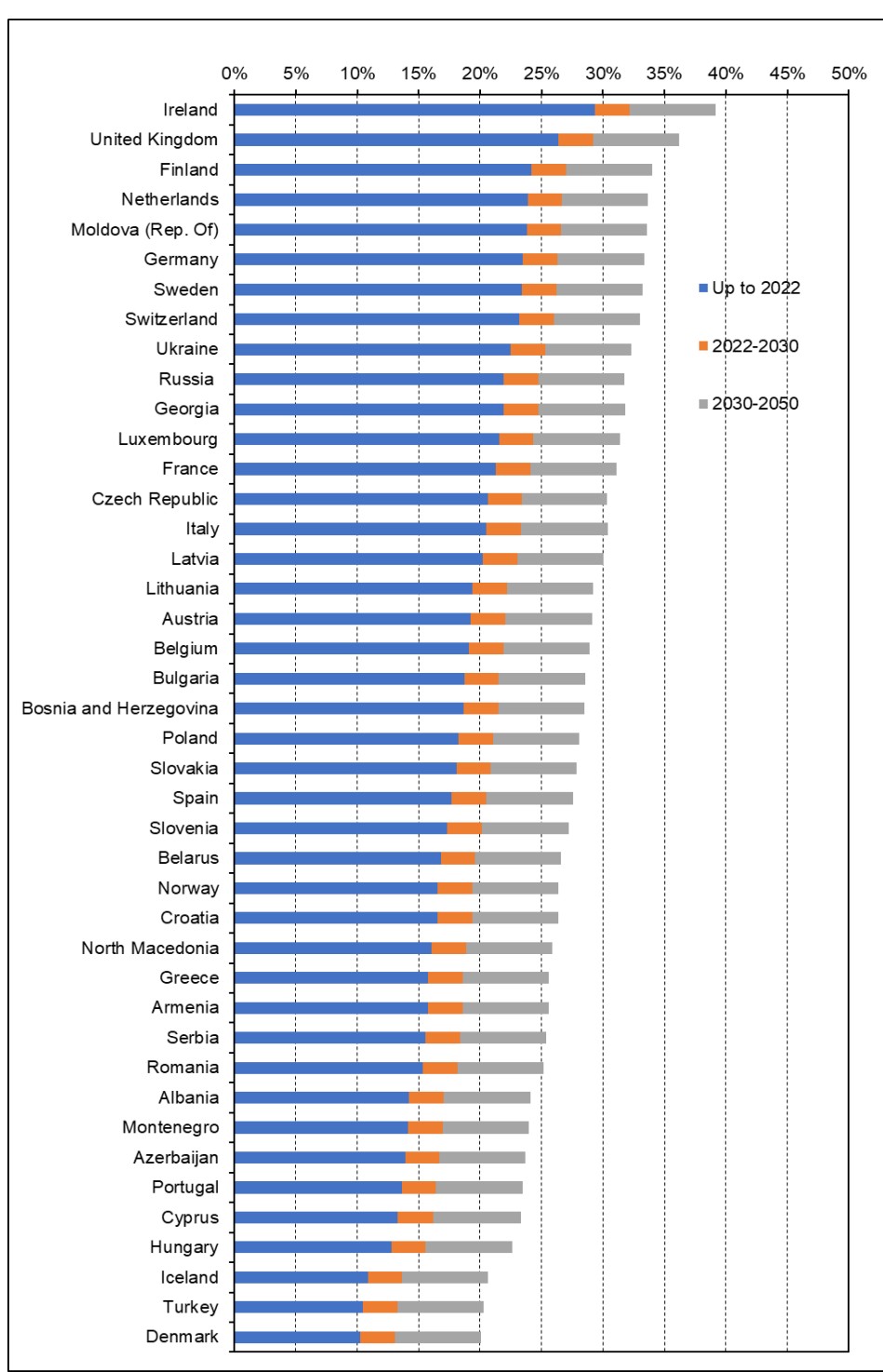

**Figure 3.** Estimated reservoir storage loss (%) by country in Europe.

*3.3. Africa*

The estimated storage loss in Africa included 2349 dams in 44 countries with a total initial storage of 702 billion m³. At present, these dams lost around 15% of their storage. By 2030 and 2050, the cumulative loss of storage is estimated to reach 17% and 24%, respectively. Nine (9) and seven (7) dams in Chad and Cape Verde, respectively, were relatively young, with their average ages of 18 and 9 years. As a result, they showed limited storage losses by 2022, which will increase after that. Seychelles is projected to lose 50% of its storage by 2050-the highest in all regions (Figure 4). Seychelles has two large

dams with cumulative storage of 1 million m$^3$ and 48 years of average age. By now (2022), Seychelles is estimated to have lost over 30% of storage in these dams. By 2050, the same loss is estimated for Seychelles, Madagascar, DR Congo, Chad, and Zambia, while another 11 countries are estimated to have lost between 25% and 30%. The lowest storage loss by 2050 is estimated for Niger (11%), with minor storage loss at present. Other countries in Africa with small storage losses (under 15%) by 2050 are Sierra Leone, Congo, Ethiopia, and Guinea. The primary reason behind this low storage loss is that they have relatively younger dams, and the sediment accumulation is still smaller than that in mature dams in the region.

North Africa has been found to have some of the lowest annual loss of storage due to sedimentation in the world, at less than 0.1% [32]. Nevertheless, the interruption of sediment flow in this region has problematic downstream effects. A clear example is the Aswan Dam on the Nile River, which has almost completely blocked sediment flow to the Nile River Delta (with the estimated trapping efficiency of the dam of 99%; Obialor et al., [33]). Based on the median LR in this study, the estimated storage losses on the Aswan dam were 18%, 21%, and 28% in 2022, 2030, and 2050, respectively.

In this study, Lesotho shows 8%, 11%, and 18% storage loss due to sedimentation by 2022, 2030, and 2050. Reservoir sedimentation is a matter of significant concern in Lesotho, providing water to itself and South Africa. A study on the Muela reservoir, with an initial reservoir capacity of 6 million m$^3$, found an average annual loss in storage capacity of 15,400 m$^3$ (around 0.26%) between 1985 to 2015 [28]. Other estimates for the same reservoir suggest an average loss of 17,500 m$^3$ [28]. Employing the median LR, the estimated annual loss of storage for the Muela reservoir in this study was around 21,000 m$^3$; thus, approximately 25% more than in the abovementioned studies.

Reservoir sedimentation, where anthropogenic activities, principally deforestation, accelerated soil erosion, has been described as a "time bomb" (Dalu et al., 2012) [34]. The Malilangwe reservoir—a dam 25.75 m high—was constructed in 1964. The reservoir was anticipated to lose in the range of 16–32% of its storage capacity within its first 100 years after the construction, depending on the future sediment yield rate [34]. Employing the median LR, the estimated storage loss of the Malilangwe reservoir up to 2022 is 20%; by 2064, it will lose 35% of its initial storage. The estimated cumulative Zimbabwean dams' storage losses for 2050 are 23% for the 247 large dams considered in this study.

The heavily dammed country in the continent, South Africa, hosts over 1250 large dams, and the estimated storage losses for the 1189 dams included in this study resulted in storage loss estimates of 18%, 21%, and 28% of the country's initial capacity for 2022, 2030, and 2050, respectively.

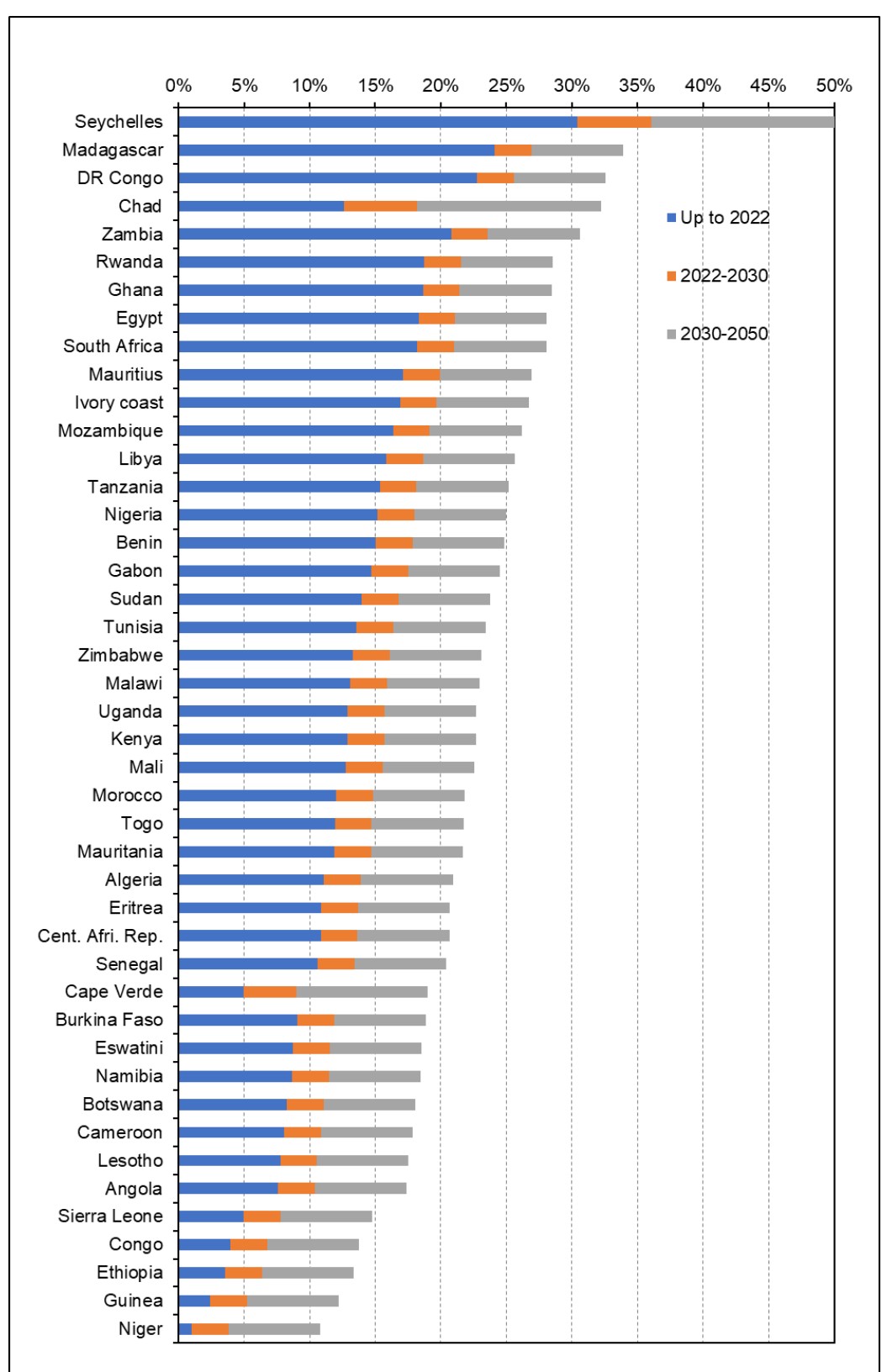

**Figure 4.** Estimated reservoir storage loss (%) by country in Africa.

### 3.4. Asia-Pacific Region

The region includes Asian countries (43), Australia, and New Zealand, totalling 45 countries with 35,252 large dams; therefore, it is the most heavily dammed region in the

world. China, India, Japan, and South Korea are among the top 10 countries in terms of the number of large dams [2]. A set of 28,045 dams (~80% of all dams in the Region on ICOLD database) had the two required parameters—construction year and initial storage capacity—and were included in the storage loss estimation. The total initial storage of these dams is ~1909 billion m$^3$, and the average dam's age of 42. By 2022, the region is estimated to have lost 13% of its initial dam storage capacity. It will lose another 10% of it by 2050 (i.e., nearly a quarter (23%) of the total initial storage capacity will be filled with sediments by mid-century). At the same time, the region remains at the lowest percentage of its initial storage loss relative to other areas (23% loss by 2050 compared to 28% in some other areas).

Japanese dams' average age is over 100 years, while Mongolian dams are the youngest, at 12 years old on average. Accordingly, as illustrated by Figure 5, loss of storage in Japan is the most significant in the region. By today (2022), it has lost 39% of its initial storage and will lose nearly 50% of initial storage by 2050. Old dams with larger capacities drive storage loss in Japan. Kantoush and Sumi [35] assessed seven large dams in the Mihi-kawa river system in Japan between 37 and 64 years old and found that those reservoirs lost from 37% to 67% of their storage capacities (an estimated 0.5% annually on average). The 3052 dams considered in this study are projected to lose 0.36% of initial storage annually from 2022 to 2050.

After Japan, Azerbaijan (24%), Israel (24%), Kazakhstan (20%), and Afghanistan (20%) have had the highest rates of storage losses till the present (2022). The exact order continues up to 2050, while Cambodia, Bhutan, Oman, Mongolia, and Brunei will have the lowest storage loss percentage of 14% by 2050. In Asia, the region where 60% of the world's population lives, water storage is crucial in sustaining water and food security. It will face a more challenging future if it loses 23% of its water storage in large dams due to sedimentation. At present (2022), Japan is the only country that has lost over 25% of its initial storage in the Asia-Pacific region. By 2050—in just 28 years from now—there will be 23 such nations in the region. This includes developed countries such as Australia, Japan, New Zealand, and South Korea, as well as the least developed countries such as Afghanistan and Bangladesh (Figure 5). Among these 23 nations are highly populated countries such as India and Bangladesh. India's Central Water Commission (2015) reported that among 141 large reservoirs over 50 years old, one-quarter had lost at least 30% of their initial storage capacity. The present study suggests that 3700 dams in India will lose 26% of their initial total storage by 2050.

China, which holds the highest number of dams in the world, has lost about 10% of its storage till the present day (2022) and will lose another 10% by 2050. In another study, Wang and Hu [36] estimated that 66% of installed storage capacity had been lost through reservoir sedimentation in China. India, China, Bangladesh, and Pakistan will be among the most populous countries in 2050 (https://www.ined.fr, accessed on 12 September 2022); their water storage will be reduced by 18%–28%.

In Iran, the hydropower Dez Dam commissioned in 1960, with initial storage of 3315 million m$^3$, has experienced close to a 20% loss in capacity through 40+ years of operation due to sedimentation [26]. This study estimates storage loss for this reservoir based on the median LR as 16% for the same period.

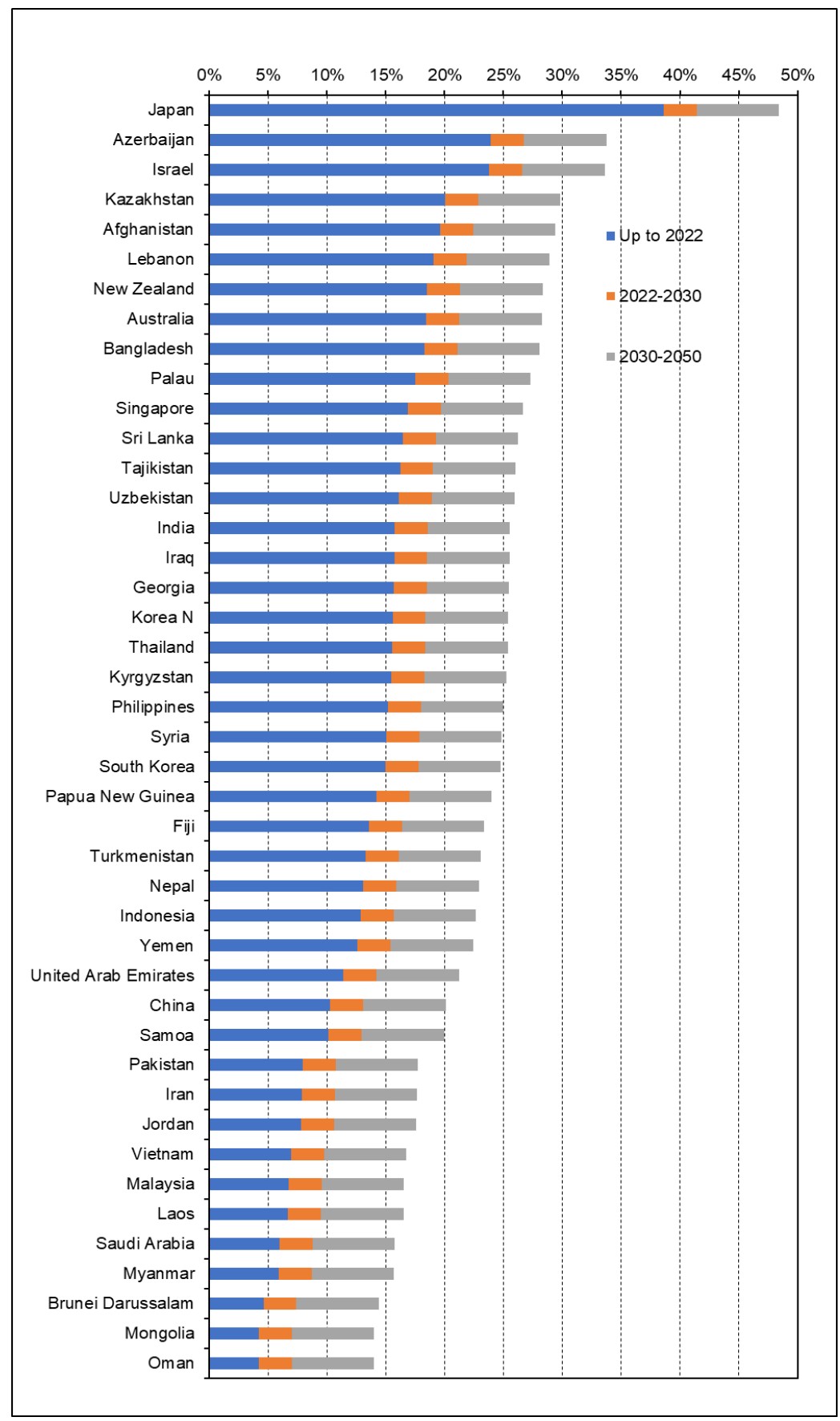

**Figure 5.** Estimated reservoir storage loss (%) by country in Asia-Pacific region.

### 3.5. Dam Capacity-Based Storage Loss Estimation

Storage loss estimation was conducted for two categories of dam sizes (under and above 1 million m$^3$) for the top ten countries with the largest number of dams: China, USA, India, Japan, Brazil, South Korea, South Africa, Canada, and Mexico. The countries included are from each major region. In total, these countries have 37,815 large dams, among them 38% (14,425), with a capacity of under 1 million m$^3$. The remaining 62% (23,490) have capacities above 1 million m$^3$ (Table 5). South Korea, Japan, and South Africa have over 60% of dams with a capacity of less than 1 million m$^3$, while over 90% of Indian and Canadian large dams are above 1 million m$^3$. Figures 6 and 7 show the statistics on storage losses for two dam size categories for the ten countries estimated for 2022, 2030, and 2050.

**Table 5.** Estimates of storage losses in 2022, 2030, and 2050 for dams under and above 1 million m$^3$.

| Country | Total Dams Considered | Dams below One Million m$^3$ of Storage | | Dams above One Million m$^3$ of Storage | |
| --- | --- | --- | --- | --- | --- |
| | | No of Dams | (%) | No of Dams | (%) |
| China | 17,681 | 8323 | 47% | 9358 | 53% |
| USA | 7469 | 1303 | 17% | 6166 | 83% |
| India | 3700 | 343 | 9% | 3357 | 91% |
| Japan | 3052 | 1881 | 62% | 1171 | 38% |
| South Korea | 1338 | 907 | 68% | 431 | 32% |
| South Africa | 1197 | 720 | 60% | 477 | 40% |
| Mexico | 1004 | 299 | 30% | 705 | 70% |
| Spain | 987 | 396 | 40% | 591 | 60% |
| Brazil | 896 | 218 | 24% | 678 | 76% |
| Canada | 491 | 35 | 7% | 456 | 93% |
| Total | 37,815 | 14,425 | 38% | 23,390 | 62% |

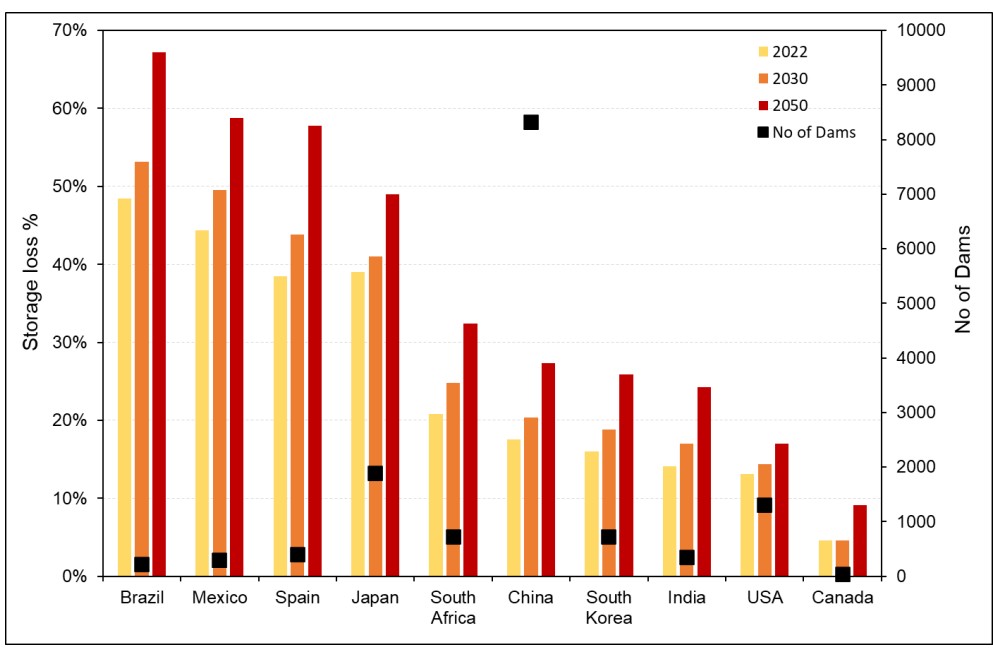

**Figure 6.** Storage loss estimates for the dams of capacity under one million m$^3$.

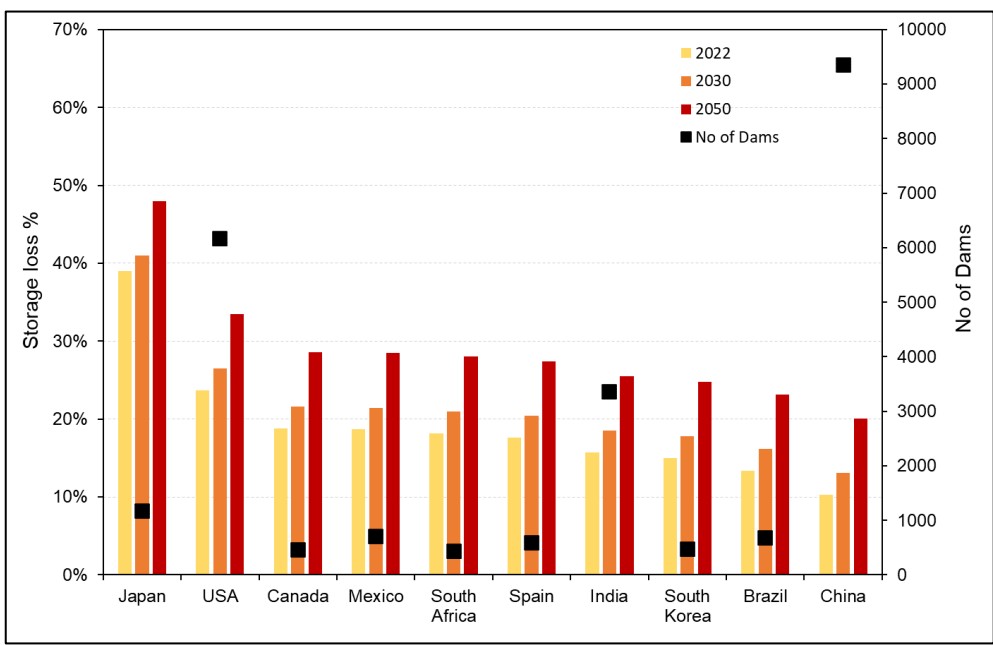

**Figure 7.** Storage loss estimates for the dams of capacity over one million m$^3$.

As Figure 6 illustrates, Brazil shows the highest level of storage losses among the ten countries in the category of dams under 1 million m$^3$, while the Canadian dams have the lowest storage losses in this category. A 19% increment in storage losses between 2022 and 2050 has been estimated for Brazil and Spain, the highest in this category, while the USA shows the lowest increment of 4%. China, the owner of the largest number of dams irrespective of capacity, shows a 10% loss increase and is positioned in the sixth among the ten countries under the category of dams with less capacity than one million m$^3$. At the same time, it is the lowest storage-loss country in the category of dams, with over 1 million m$^3$ dams. In the category of dams over 1 million m$^3$, as shown in Figure 7, Japan appears to have the highest percentage of storage loss by 2050, and most countries in this category show about a 10% increment of losses between 2022 and 2050. Overall the category below 1 million m$^3$ demonstrates more significant storage losses than the one over 1 million m$^3$. However, as evident from the LRs employed, as explained in Table 1, Wisser et al. [10] concluded that dams with smaller storage have higher storage losses due to sedimentation. The illustrations above support and reflect that conclusion.

## 4. Conclusions

The study estimated storage losses of large dams due to sedimentation at the global scale, by country. It utilized the methodology suggested by Wisser et al. [10], who derived several annual storage loss rates that were potentially applicable globally, based on the analysis of numerous reservoir bathymetry studies from around the world and applied those rates to 6399 reservoirs globally.

The current study applied the method to the much larger and most comprehensive global dataset of large dams available at present, that of the International Commission on Large dams (ICOLD). Out of almost 59,000 dams on the ICOLD database, 47,403 dams were used in the analysis, those with the two key parameters, initial storage capacity and the first year of dam exploitation. Over 85% of all large dams in the database are in 150 countries of the four main geographic regions—the Americas (North, Central, and South), Europe, Africa, and Asia-Pacific (including Australia and New Zealand).

The estimates of storage losses by country were made for several time horizons: 2022, 2030, and 2050. The first corresponds to the present time, the second corresponds to the end of the decade and the end of the current period of the Sustainable Development Goals (SDGs), and the last corresponds to the middle of the century. All time horizons

are either current or near future. Estimated storage losses are cumulative, i.e., they build up progressively from "initial storage" to 2022, next to 2030, and then to 2050 eventually. Initial storage is estimated as a sum of the capacities of all storage dams in each country during the first year of exploitation of a dam. The attempt was made to compare, where possible, storage loss estimates with recorded storage losses identified in the literature.

It is shown that the regionwide storage loss percentages for 2022 range between major regions between 13% and 19%; for 2030, between 16% and 21%; and for 2050, between 23% and 28%. Globally, storage losses by now have reached 16% of the initial storage capacity. By 2050, storage losses are estimated to be the lowest (23%) in the Asia-Pacific region, while in the Americas and Europe, storage losses are the highest (28%). Globally, an additional 10% of storage loss will likely occur from 2022 (16%) to 2050 (26%), with averaged annual losses of 0.36% of initial global storage capacity.

In the 19 countries of the Americas, a cumulative initial storage capacity of 2810 billion m$^3$ is estimated to decline to 2014 billion m$^3$ by 2050 (a decrease of 28%). Panama will endure the most significant storage loss of 38% of its storage by 2050, and the USA will see the second largest (34%). Uruguay, Suriname, and Canada are estimated to lose 33%, 30%, and 29% of their initial storage capacity, respectively, by 2050. At the "tail end" of storage losses in the Americas are Bolivia (21%), Colombia (19%), and Belize (16%). Currently, only Panama in the Americas has lost over 25% of initial storage. By 2030, USA and Uruguay will reach the same level of losses, and by 2050, 10 out of 19 countries in the Americas will lose over 25% of their initial storage.

In Europe, the total initial storage of 895 billion m$^3$ has already declined by 19% and will further reduce by 28% by 2050. Among the 42 European countries considered, 33 will likely lose over 25% of initial storage by 2050, doubling its 2022 number. Ireland has the highest level of storage loss at all time horizons (39% by 2050), followed by the UK, Finland, the Netherlands, and Moldova, while Denmark has the lowest (20%). Turkey, Iceland, Hungary, and Cyprus appear to be the other least-impacted countries in Europe.

In 44 African countries, the total initial storage of 702 billion m$^3$ has declined by 15% by 2022. By 2030 and 2050, the cumulative loss of storage is estimated to reach 17% and 24%, respectively. By 2022, over 30% of storage loss is estimated for Seychelles, Madagascar, DR Congo, Chad, and Zambia, while another 11 countries are estimated to have lost between 25% and 30%. The lowest storage loss by 2050 is estimated for Niger (11%), and other countries in Africa with small storage losses (under 15%) by 2050 are Sierra Leone, Congo, Ethiopia, and Guinea.

In 45 Asian and Pacific countries, with almost 80% of all its large dams considered in this study, the total initial storage of 1909 billion m$^3$ is estimated to have already declined by 13% by 2022. By 2050, the region will likely lose another 10% of its storage, and the cumulative loss will reach nearly a quarter (23%) of the total initial storage capacity. At the same time, the region remains at the lowest percentage of its initial storage loss relative to other areas. The dams in Japan are, on average, over 100 years, while Mongolian dams are the youngest—12 years old on average. Accordingly, loss of storage in Japan is the most significant in the region. By today (2022), it lost 39% of its initial storage and will lose nearly 50% of its initial storage by 2050. After Japan, Azerbaijan (24%), Israel (24%), Kazakhstan (20%), and Afghanistan (20%) have had the highest rates of storage losses till the present (2022). This exact order continues up to 2050, while Cambodia, Bhutan, Oman, Mongolia, and Brunei will have the lowest storage loss percentage of 14% by 2050.

Obviously, the application of a reservoir capacity-based uniform storage loss rate is a rather simplified approach and hence a limitation of this study. A reservoir's actual storage loss rate depends on spatial and temporal factors such as land use/land cover and climate variability at the reservoir's location. A more accurate estimation of storage losses due to sedimentation can be achieved through consistent basin-wide monitoring of sediment transport and frequent bathymetry surveys of individual reservoirs. Such data need to be built up over time. In the meantime, an attempt was made to minimize the above limitation by identifying the ""most likely"" uniform LR; this was executed by comparing estimated

storage losses with the reported storage losses for 247 dams. Such generalizations and simplifications are likely, inevitable in global studies such as this one.

The study ranked all countries in each region in descending order by the magnitude (percentage) of their cumulative storage losses by 2050 (storage losses at other time horizons generally follow the same pattern as 2050). These illustrations are not to show "losers" or "winners" but to illustrate the relative importance of the problem for various nations. New dam construction continues in some regions, but the rate of new dam construction has declined drastically in the last four decades [2] from some 1000 new dams per year in the middle of the 20th century to only some 50 per year at present. This is more than an order of magnitude decrease in new large dam development. It is, therefore, unlikely that with such a dramatic (and continuing) reduction in new dam construction, storage losses to sedimentation may be offset. However, a more detailed assessment of this on a regional or country basis may be required.

Numerous innovative solutions have been developed and are applied worldwide to manage reservoir sedimentation. Dredging is one of those, but it can be costly and, eventually, only a temporary solution, with an ongoing expense. Sediment flushing is more financially attractive but may have significant adverse impacts downstream [34]. Solutions such as bypass (or diversion) may be seen as "near-nature" and are gaining traction due to the growing public quest to minimize dams' adverse environmental impacts. Bypassing is a technique that diverts the flow downstream via a separate channel, often used to manage high-flow events where sediment concentration is particularly high [37]. Bypass tunnels can achieve an efficiency of 80%–90% of all sediment at their optimum operational levels [38]. Enhancement of the dam height is another alternative to recover the storage loss due to sedimentation. However, this should be executed only after a careful assessment of the structural strength of the dam. The benefits of doing this are manifold: more storage for the intended dam function. However, the increase in height means that the reservoir area will also expand, potentially displacing communities and leading to certain habitat losses. Complete removal of dams, including those that are filled with sediments, is also a (slowly) emerging practice. Dam removal can bring rivers back to their natural state and reestablish the natural river sediment transport. However, certain treatment and disposal of sediments accumulated in a reservoir may be needed as they may contain heavy metals and other toxins.

Clearly, this study's results need to be interpreted at the local level considering local specifics and factors beyond this study's scope. However, what is important to underline is that the overall magnitude of water storage losses due to sedimentation is quite disturbing. It adds to the list of water development issues that the world is already facing and has been unable to resolve.

**Author Contributions:** Conceptualization, D.P. and V.S.; literature survey, S.W.; methodology, D.P.; analysis, D.P.; writing—original draft preparation, D.P. and S.W.; writing—review and editing, V.S. All authors have read and agreed to the published version of the manuscript.

**Funding:** This research is supported by the funds received by UNU-INWEH through the long-term agreement with Global Affairs Canada.

**Institutional Review Board Statement:** Not Applicable.

**Informed Consent Statement:** Not Applicable.

**Data Availability Statement:** Not Applicable.

**Acknowledgments:** The authors are grateful to anonymous reviewers for their helpful and constructive comments on the manuscript. Thanks are due to Jonathon Kusa and Kristen Coveleski at Inter-Fluve, Inc. USA, for their valuable insights at the initial stages of this study. The authors' appreciation also goes to Angelina Abi Daoud at McMaster University for her initial literature survey on the estimation techniques of dam sedimentation.

**Conflicts of Interest:** The authors declare no conflict of interest.

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
