# Peer review of "Present and Future Losses of Storage in Large Reservoirs Due to Sedimentation: A Country-Wise Global Assessment"

_sustainability, doi:10.3390/su15010219_

Round 1

Reviewer 1 Report

The paper presents a study of global variation of storage losses of large dams due to sedimentation. It is a suitable topic for Sustainability MDPI journal. The manuscript is professionally written, clear, and easy to read. Furthermore, it is of great interest and importance to the research field. Therefore, I would recommend a revision following my comments below.

  • The authors should improve the Methodology.
  • The manuscript's novelty should be (more) explored in the Abstract, Introduction, and Conclusions.
  • Figure 1 is of low resolution. Furthermore, why is there a square hole inside Brazil? 

Author Response

01st  December 2022

Dear Reviewer-1,

Thank you very much for reviewing our manuscript and providing constructive comments to improve. we tried our best to address your comments, and our answers/revisions to your comments are attached.

Yours sincerely,

Reviewer 2 Report

Review of the manuscript “Present and future losses of storage in large reservoirs due to 2 sedimentation: a country-wise global assessment”. The manuscript estimates the cumulative reservoir storage losses at the global scale for the present and future. To this end, the authors used previously observed successive bathymetry survey data from 1024 dam data for their assessment. It’s intriguing research which perfectly aligns with the sustainability scope. Unfortunately, the method does not seem convincing to address such a complex sedimentation phenomenon. The sedimentation rate changes instantaneously depending upon the nature of the sediment load. There must be a qualified model application such as Telemac2D to confirm the observed data for at least a single basin depending upon the available model data. The artwork is appropriate and legible. However, it is simple and shows global maps but is unable to support any solid evidence. I will recommend a major revision providing an opportunity to the authors for including an acceptable model for sediment load calculation.

Author Response

01st  December 2022

Dear Reviewer-2,

Thank you very much for reviewing our manuscript and providing constructive comments to improve. we tried our best to address your comments, and our answers/revisions to your comments are attached.

Yours sincerely,

Reviewer 3 Report

The study estimated storage losses of large dams due to sedimentation for each country globally. There are a few points that the authors should improve.

First, in Line 151-152, the authors claimed that median LR was found to correlate best with documented estimates in terms of Root Mean Square Error and Nash-Sutcliffe efficiency coefficient. Please clarify how the two metrics are calculated add a table/figure to show the values of the two metrics using different LR to prove your statement.

Second, since the authors estimated the storage loss at present day 2022, it worth doing by comparing your estimated storage loss with actual observed storage loss to justify your finding. If the global level present-day data is not available, using a few reservoirs for comparison can also be a good justification.

Third, did the authors consider the rate of constructing new dams? It should be interesting to see how the new constructions offset the sediment-induced storage loss.

Fourth, other than country-wise analysis, it should be interesting to add a comparison figure of storage loss between reservoirs > 1 Mm3 and < 1Mm3 to see the influence of sedimentation to different reservoir sizes.

Author Response

01st  December 2022

Dear Reviewer-3,

Thank you very much for reviewing our manuscript and providing constructive comments to improve. we tried our best to address your comments, and our answers/revisions to your comments are attached.

Yours sincerely,

Round 2

Reviewer 2 Report

Review of the manuscript “Present and future losses of storage in large reservoirs due to sedimentation: a country-wise global assessment” 

The authors have addressed my raised issue regarding methodology (model application) in a somehow justified manner. Unfortunately, this study still seems an improvised version of Wisser et al., 2013 (the authors mentioned) in which the authors have just updated the dataset being restricted to only storage rather than population number and river basins. 

My second concern was the variability of sedimentation loss rate (LR), Wisser et al., 2013 used a uniform but there is no information regarding the uniformity of LR in the present study. It is the major limitation of this study. 

Based on the above-mentioned facts, I would suggest making a section on the limitations of the study. Furthermore, emphasize the methods of sediment removal e.g dredging, flushing and enhancement of storage capacity (dam wall enhancement). What is the difference between conventional and modern dams? 

Author Response

Dear Reviewer,

Thank you very much for your comments and suggestion to improve our manuscript. We addressed them to the best of our knowledge and understanding. 

Please check the attached PDF for our detailed reply to your comments and revisions.

Best regards,

Duminda Perera
